# On the Robustness of Reward Models for Language Model Alignment

Jiwoo Hong [1]   Noah Lee [1]   Eunki Kim [1]   Guijin Son [2]   Woojin Chung [1]

Aman Gupta [3]   Shao Tang [3]   James Thorne [1]

## Abstract

The Bradley-Terry (BT) model is widely practiced in reward modeling for reinforcement learning with human feedback (RLHF). Despite its effectiveness, reward models (RMs) trained with BT model loss are prone to *over-optimization*, losing generalizability to unseen input distributions. In this paper, we study the cause of over-optimization in RM training and its downstream effects on the RLHF procedure, accentuating the importance of distributional robustness of RMs in unseen data. First, we show that the excessive dispersion of hidden state norms is the main source of over-optimization. Then, we propose batch-wise sum-to-zero regularization (**BSR**) to enforce zero-centered reward sum per batch, constraining the rewards with extreme magnitudes. We assess the impact of BSR in improving robustness in RMs through four scenarios of over-optimization, where BSR consistently manifests better robustness. Subsequently, we compare the plain BT model and BSR on RLHF training and empirically show that robust RMs better align the policy to the gold preference model. Finally, we apply BSR to high-quality data and models, which surpasses state-of-the-art RMs in the 8B scale by adding more than 5% in complex preference prediction tasks. By conducting RLOO training with 8B RMs, AlpacaEval 2.0 reduces generation length by 40% while adding a 7% increase in win rate, further highlighting that robustness in RMs induces robustness in RLHF training. We release the code, data, and models: `https://github.com/LinkedIn-XFACT/RM-Robustness`.

[1]KAIST AI [2]OneLineAI [3]LinkedIn Corporation. Correspondence to: Jiwoo Hong <jiwoo_hong@kaist.ac.kr>.

*Proceedings of the 42$^{nd}$ International Conference on Machine Learning*, Vancouver, Canada. PMLR 267, 2025. Copyright 2025 by the author(s).

## 1. Introduction

Reward models (RMs) are crucial components in reinforcement learning with human feedback (RLHF), being *proxies* for human preference for aligning large language models (LLMs) (Christiano et al., 2017; Ziegler et al., 2020; Casper et al., 2023; Wang et al., 2024a). RMs map text sequences to a scalar score, and are incorporated into the RLHF pipeline by directly using the scores (Ziegler et al., 2020; Ahmadian et al., 2024), or labeling pairwise preferences (Rafailov et al., 2023; Hong et al., 2024; Meng et al., 2024).

The Bradley-Terry (BT) model (Bradley & Terry, 1952) formulation is widely adopted to train RMs using a tuple of a prompt and two corresponding responses with a pre-annotated preference relation (*e.g.*, chosen and rejected), maximizing the margin of rewards between the responses. Recent works proposed *additive approaches* to the BT model to further adjust it to the neural language model context (Yuan et al., 2024; Coste et al., 2024; Yang et al., 2024b). However, these do not adjust the learning objective of the BT model at a fundamental level.

Reward model over-optimization, where RMs overfit to the train set and eventually losing the alignment capability to the true preference distribution, is a typical limitation of neural RMs (Gao et al., 2023; Coste et al., 2024). Efforts to curate high-quality pairwise preference datasets has contributed to enhancing RMs (Yuan et al., 2024; Liu et al., 2024a; Cui et al., 2025), assessed through pairwise preference benchmarks (Kim et al., 2024; Lambert et al., 2025; Liu et al., 2025b). However, using LLMs either as preference data generators or annotators often induces inherent biases like verbosity bias and self-enhancement biases (Saito et al., 2023; Zhang et al., 2024b; Chen et al., 2024a) and obstructs understanding of underlying RM over-optimization.

In this paper, we show that reward modeling with the BT model induces *excessive hidden state norm dispersion*, leading to the over-optimization problem. We propose batch-wise sum-to-zero regularization (**BSR**) as a straightforward solution to control the dispersion by penalizing abnormal reward outliers, eventually reducing the dispersion in hidden state norms. We categorize four generalization scenarios for RMs with respect to prompt and response space, allowing

fine-grained analysis of over-optimization and robustness. Based on these scenarios, we propose a regularized reward modeling objective $\mathcal{L}_{\text{BT-BSR}}$, excelling baseline methods in all generalization scenarios and improving accuracy in complex tasks over 5% in RM-Bench compared to simple BT model. Further analysis of the propagation of robustness in RMs to RLHF training shows **BSR** yields a 40% decrease in generation length with 7% gain in AlpacaEval 2.0.

## 2. Background

### 2.1. Preliminaries

**Reward modeling with language models**   Reward models (RMs) for language model alignment use the Bradley-Terry (BT) model (Bradley & Terry, 1952) for human preference between chosen and rejected responses $y_w$ and $y_l$ given a prompt $x$ as:

$$P(y_w \succ y_l|x) = \frac{e^{r(x,y_w)}}{e^{r(x,y_w)} + e^{r(x,y_l)}}. \quad (1)$$

Typically, they are implemented as classifiers with projection head $W_p$ (Ziegler et al., 2020; Ouyang et al., 2022):

$$r(x,y) = W_p^T \cdot h(x,y), \quad W_p \in \mathbb{R}^{H \times 1}, \quad (2)$$

$h(x,y)$ refers to the last hidden state from the backbone language model with the hidden dimension of $H$, given the prompt and response pair $(x,y)$. In practice, both encoder-only (Jiang et al., 2023b) and decoder-only language models (Yuan et al., 2024; Liu et al., 2024a) are used as a backbone model that returns $h(x,y)$. And $W_p$ is the projection head which the weights are randomly initialized by $\mathcal{N}(0, (H + 1)^{-1})$ (Stiennon et al., 2020; Huang et al., 2024).

RMs are fine-tuned to maximize the margin between the chosen and rejected responses' scores (Christiano et al., 2017; Stiennon et al., 2020):

$$\mathcal{L}_{\text{BT}} = -\mathbb{E}_{(x,y_w,y_l) \sim \mathcal{D}} \left[ \log \frac{e^{r(x,y_w)}}{e^{r(x,y_w)} + e^{r(x,y_l)}} \right], \quad (3)$$

where $\mathcal{D}$ is the set of triplets comprising chosen and rejected responses $y_w$ and $y_l$ given the fixed prompt $x$, and $r(x,y) \in \mathbb{R}$ is a score assigned to $(x,y)$.

**Preference modeling with BT model**   BT model is defined for *directly comparable* components (Bradley & Terry, 1952; Davidson, 1970; Huang et al., 2004), leaving the responses $y_{i,j} \sim \mu(\cdot|x_i)$ given the fixed prompt $x_i$ to be comparable. For each prompt $x_i$, there exists a BT model with prompt-specific parameters $\phi_i$ that defines the preference:

$$P(y_{i,1} \succ y_{i,2}|x_i; \phi_i) = \frac{e^{s_{\phi_i}(x_i,y_{i,1})}}{e^{s_{\phi_i}(x_i,y_{i,1})} + e^{s_{\phi_i}(x_i,y_{i,2})}}, \quad (4)$$

where $s_{\phi_i}(x_i, \cdot)$ represents the scoring function specific to prompt $x_i$. However, the reward model $r_\theta$ parameterized by the language model unifies these prompt-specific BT models by learning a single parameterization $\theta$ that maps the entire space of prompts and responses $\mathcal{X} \times \mathcal{Y}$ to preference scores.

**Reward model parameterization**   Let $\mathcal{M} = M_1, ..., M_K$ be a set of language models where each $M_k$ defines a conditional distribution over responses given a prompt. For any prompt $x$ in the prompt set $\mathcal{X} = \{x_i\}_{i=1}^N$, each model $M_k$ induces a prompt-specific response space $\mathcal{Y}_k(x) = \text{supp}(M_k(\cdot|x))$:

$$\mathcal{Y}(x) = \bigcup_{k=1}^K \mathcal{Y}_k(x), \quad (5)$$

where $\mathcal{Y}(x)$ represents all possible responses for prompt $x$ across all models. Thus, $r_\theta$ can be represented as:

$$r_\theta : \mathcal{X} \times \mathcal{Y}(x) \to \mathbb{R}, \quad (6)$$

$$\text{where} \quad s_{\phi_i}(x_i, \cdot) = r_\theta(x_i, \cdot) \quad \forall x_i \in \mathcal{X}. \quad (7)$$

This shared parameterization enables learning preference patterns that generalize across the prompt space while maintaining the BT structure within each prompt context.

### 2.2. Problem setup: robustness of reward models

Motivated by how RMs parameterize multiple prompt-level BT models into a single parameter $\theta$, we expand the issue of *reward model over-optimization* (Gao et al., 2023) by categorizing generalization scenarios based on prompt disjointness and response disjointness. By doing so, **we study how each generalization scenario of RM is affected by over-optimization and how to improve the robustness**.

We follow how Gao et al. (2023) sets the synthetic gold RM instead of human annotations for controlled assessment. Let there exist a true preference model $r^* : \mathcal{X} \times \mathcal{Y}(x) \to \mathbb{R}$. Let $\mathcal{M}_{\text{train}} \subset \mathcal{M}$ and $\mathcal{M}_{\text{valid}} \subset \mathcal{M}$ be the sets of models used for generating responses in the training and validation sets respectively. The corresponding response spaces are:

$$\mathcal{Y}_{\text{train}}(x) = \bigcup_{M_k \in \mathcal{M}_{\text{train}}} \mathcal{Y}_k(x) \quad (8)$$

$$\mathcal{Y}_{\text{valid}}(x) = \bigcup_{M_k \in \mathcal{M}_{\text{valid}}} \mathcal{Y}_k(x). \quad (9)$$

Thus, the train set $\mathcal{D}_{\text{train}}$ for training $r_\theta$ can be defined as:

$$\mathcal{D}_{\text{train}} = \{(x, y_w, y_l)|(y_w, y_l) \sim \mathcal{Y}_{\text{train}}(x), x \in \mathcal{X}_{\text{train}}\}. \quad (10)$$

First, *prompt disjointness* is defined as two prompt sets $\mathcal{X}_{\text{train}}$ and $\mathcal{X}_{\text{valid}}$ being disjoint. And *response disjointness* is defined as two response spaces $\mathcal{Y}_{\text{train}}$ and $\mathcal{Y}_{\text{valid}}$ having disjoint response model sets $\mathcal{M}_{\text{train}}$ and $\mathcal{M}_{\text{valid}}$. From this context, we categorize RM generalization scenarios into:

1. **In-domain ($\mathcal{D}_{\text{ID}}$):** $\mathcal{D}_{\text{ID}}$ shares the same prompts with $\mathcal{D}_{\text{train}}$ and responses are sampled from the same response model set $\mathcal{M}_{\text{train}}$:

$$\{(x, y_w, y_l)|(y_w, y_l) \sim \mathcal{Y}_{\text{train}}(x), x \in \mathcal{X}_{\text{train}}\}. \quad (11)$$

2. **Prompt-disjoint ($\mathcal{D}_{\sim\text{Prompt}}$):** $r_\theta$ being generalizable to the unseen prompt set $\mathcal{X}_{\text{valid}}$ but with the same response model set $\mathcal{M}_{\text{train}}$:

$$\{(x, y_w, y_l)|(y_w, y_l) \sim \mathcal{Y}_{\text{train}}(x), x \in \mathcal{X}_{\text{valid}}\}. \quad (12)$$

3. **Response-disjoint ($\mathcal{D}_{\sim\text{Response}}$):** $r_\theta$ being generalizable to the seen prompt set $\mathcal{X}_{\text{train}}$ but with the unseen response model set $\mathcal{M}_{\text{valid}}$:

$$\{(x, y_w, y_l)|(y_w, y_l) \sim \mathcal{Y}_{\text{valid}}(x), x \in \mathcal{X}_{\text{train}}\}. \quad (13)$$

4. **Mutual-disjoint ($\mathcal{D}_{\sim\text{Mutual}}$):** $r_\theta$ being generalizable to both unseen prompts $\mathcal{X}_{\text{valid}}$ and response models $\mathcal{M}_{\text{valid}}$:

$$\{(x, y_w, y_l)|(y_w, y_l) \sim \mathcal{Y}_{\text{valid}}(x), x \in \mathcal{X}_{\text{valid}}\}. \quad (14)$$

From this context, we define RM over-optimization as RM's accuracy on $\mathcal{D}_{\text{train}}$ and $\mathcal{D}_{\text{ID}}$ increasing, while its performance on $\mathcal{D}_{\sim\text{Prompt}}$, $\mathcal{D}_{\sim\text{Response}}$, and $\mathcal{D}_{\sim\text{Mutual}}$ stagnate or degrades. Intuitively, it is equivalent to $r_\theta$ losing its alignment with $r^*$ in the general cases after training with limited samples.

### 2.3. Post-analysis: Reinforcement learning with human feedback

We then analyze the propagation of over-optimization in the downstream RLHF applications. By fine-tuning the language model $\pi$ with reinforcement learning (RL) algorithms to maximize the reward with respect to $r_\theta$ as human preference proxies, the policy $\pi$ is trained to maximize the average reward for its response $y \sim \pi(\cdot|x)$ given the prompt $x \sim \mathcal{D}$:

$$\max_\pi \mathbb{E}_{y\sim\pi(\cdot|x), x\sim\mathcal{D}} [r_\theta(x, y)] - \beta \mathbb{D}_{\text{KL}} (\pi(y|x)|\pi_{\text{ref}}(y|x)). \quad (15)$$

To understand the impact of over-optimization in RLHF, we assess the gold reward scores of interim $\pi$ during the training. Doing so, **we verify if maximizing $r(x, y)$ is aligned with maximizing $r^*(x, y)$.**

## 3. Experiments

We set ArmoRM (Wang et al., 2024b) as the gold preference model $r^*$, a setup similar to that of Gao et al. (2023). We selected ArmoRM as it utilizes a wide range of representative synthetic preference data and is reported to be robust to various biases (Wang et al., 2024b; Meng et al., 2024).

### 3.1. Part 1: Reward model over-optimization

**Models** We use two different model families, Llama-3 (Dubey et al., 2024) and Qwen2.5 (Yang et al., 2024a), with varying sizes. We select Llama-3.2-1B, 3B, and Llama-3.1-8B base models from the Llama-3, along with Qwen2.5-1.5B, 3B, and 7B base models from the Qwen2.5. We use UltraChat (Ding et al., 2023, UC) to conduct supervised fine-tuning (SFT) for every model. Detailed SFT and reward modeling training configurations are in Appendix A.

**Datasets** We adopt UltraFeedback (Cui et al., 2025, UF), which harnesses 17 different models with varying model families to sample four responses per prompt, to set one train set and four validation sets as described in Section 2.2.

We set the original 17 models as $\mathcal{M}_{\text{train}}$ and select four disjoint models as $\mathcal{M}_{\text{valid}}$: Gemma-2-2B-It (Team et al., 2024), Olmo2-7B-Instruct(OLMo et al., 2025), SmolLM2-1.7B-Instruct (Allal et al., 2025), and Mistral-Instruct-v0.2 (Jiang et al., 2023a). We excluded the Llama and Qwen families to avoid contamination with the training models and $\mathcal{M}_{\text{train}}$. After generating four more responses per prompt with them, we prepare a train set $\mathcal{D}_{\text{train}}$ with 51,200 rows and four validation sets: $\mathcal{D}_{\text{ID}}, \mathcal{D}_{\sim\text{Prompt}}, \mathcal{D}_{\sim\text{Response}}$, and $\mathcal{D}_{\sim\text{Mutual}}$. We report detailed preprocessing procedures in Appendix B.

### 3.2. Part 2: Propagation of RM robustness in RLHF

We simulate the impact of robustness in RM during reinforcement learning with human feedback (RLHF) with RLOO (Ahmadian et al., 2024). We test if the trained model with each RM can align with the gold preference model $r^*$.

**Models and dataset** We set the Qwen2.5-1.5B base model with supervised fine-tuning (SFT) applied using UC as an initial policy. We employ Qwen2.5-3B based RMs trained with $\mathcal{L}_{\text{BT}}$ and $\mathcal{L}_{\text{BT-BSR}}$ from Section 3.1, as they demonstrated the highest overall performance in Figure 5 for both methods. We use $\mathcal{X}_{\text{valid}}$ as the seed prompts for experiments. We report the additional training configurations in Table 4.

### 3.3. Part 3: Real-world impact of robustness in RM

Finally, we extend our experiments in Sections 3.1 and 3.2 to the 8B model and high-quality synthetic preference data to demonstrate the scalability and effectiveness of the proposed method.

**Models and datasets** We use TULU3 SFT mixture (Lambert et al., 2024) to conduct SFT on Qwen2.5-1.5B. Then, we employ Llama-3.1-8B based RMs trained with $\mathcal{L}_{\text{BT}}$ and $\mathcal{L}_{\text{BT-BSR}}$ on Skywork-Reward-Preference-80K-v0.2[1], high-quality synthetic preference dataset (Liu et al., 2024a; 2025b). We use the same $\mathcal{X}_{\text{valid}}$ as the seed prompts for RLOO. We report the additional training configurations in Table 4.

---

[1] https://huggingface.co/datasets/Skywork/Skywork-Reward-Preference-80K-v0.2

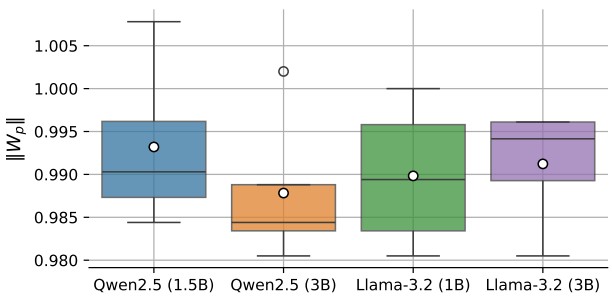

Figure 1. $||W_p||$ distribution after reward modeling for four seeds each. $||W_p||$ generally stays around one after the training.

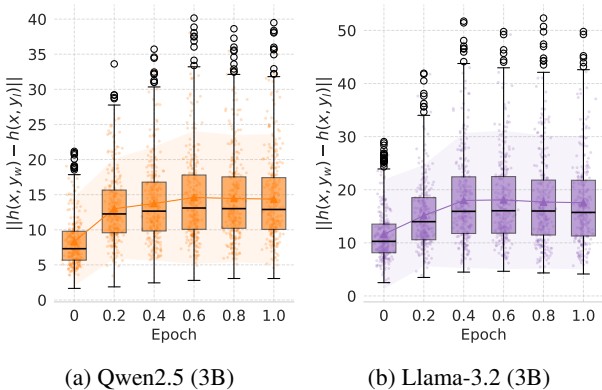

(a) Qwen2.5 (3B)  (b) Llama-3.2 (3B)

Figure 2. Growth of $||h(x, y_w) - h(x, y_l)||$ throughout reward modeling with $\mathcal{L}_{\text{BT}}$. The variance of the hidden state difference grows incrementally with the right-skewed distribution. The width of the colored area indicates the standard deviation.

## 4. Robustness and Over-optimization in RMs

In this section, we propose excessive dispersion in the hidden states of the reward model (RM) as a cause of over-optimization. We support our point by analyzing the training dynamics of $\mathcal{L}_{\text{BT}}$ in Section 4.1. Then, we propose *batch-wise sum-to-zero regularization* (BSR) as a method to mitigate such dispersion, enhancing the robustness of RMs.

### 4.1. Hypothesis

Given the prompt and response pair $(x, y)$, the final score of RM $r_\theta$ as a dot product between the projection head $W_p$ and the hidden state $h(x, y)$ can be decomposed into:

$$r_\theta(x, y) = ||W_p|| \cdot ||h(x, y)|| \cdot \cos\psi, \quad (16)$$

where $\cos\psi$ represents the cosine similarity between the two vectors $W_p \in \mathbb{R}^{H \times 1}$ and $h(x, y) \in \mathbb{R}^{H \times 1}$. Typically, the norm of two vectors $||W_p|| \cdot ||h(x, y)||$ largely contributes to maximize the softmax value and cause over-confidence issue (Wei et al., 2022), especially in the context of large language models (LLMs) that have large hidden sizes (Yang et al., 2024a; Dubey et al., 2024). Inspired by Wei et al. (2022), we hypothesize that excessive growth of hidden state norm dispersion is a major cause of RM over-optimization.

**Chosen and rejected responses share projection head**
Within three components in Equation (16), we first analyze the contribution of $W_p$ to over-optimization. Let $\Delta r$ the reward margin $r(x, y_w) - r(x, y_l)$ for the prompt $x$ and chosen and rejected responses $y_w$ and $y_l$:

$$\Delta r = W_p^T (h(x, y_w) - h(x, y_l)). \quad (17)$$

As Equation (3) is equivalent to:

$$\mathcal{L}_{\text{BT}} = -\mathbb{E}_{(x, y_w, y_l) \sim \mathcal{D}} [\log\sigma(\Delta r)], \quad (18)$$

the gradient updates for $W_p$ can be written as:

$$\frac{\partial\mathcal{L}_{\text{BT}}}{\partial W_p} = -(1 - \sigma(\Delta r)) \cdot \frac{\partial\Delta r}{\partial W_p} \quad (19)$$

$$\frac{\partial\Delta r}{\partial W_p} = h(x, y_w) - h(x, y_l), \quad (20)$$

where $\sigma(x) = (1 + \exp(-x))^{-1}$ is the sigmoid function. As $\sigma(-x) = 1 - \sigma(x)$,

$$\frac{\partial\mathcal{L}_{\text{BT}}}{\partial W_p} = -\sigma(-\Delta r) \cdot (h(x, y_w) - h(x, y_l)). \quad (21)$$

Then, the gradient norm for $W_p$ is:

$$\left|\left|\frac{\partial\mathcal{L}_{\text{BT}}}{\partial W_p}\right|\right| = \sigma(-\Delta r) \cdot ||h(x, y_w) - h(x, y_l)||. \quad (22)$$

Having $\sigma(-\Delta r)$ and $||h(x, y_w) - h(x, y_l)||$ as main components, the gradient norm of $W_p$ saturates as $\mathcal{L}_{\text{BT}}$ is minimized by maximizing $\Delta r$. Despite $\Delta r \simeq 0$ in the initial phase of training, $||h(x, y_w) - h(x, y_l)||$ is minimal at the beginning as the output hidden states of large language models (LLMs) (*i.e.*, backbone LM of RMs) have low effective rank, being concentrated to certain regions (Biś et al., 2021; Wei et al., 2024).

Thus, $||W_p||$ is expected to have marginal difference with its initial value, which is $\mathbb{E}[||W_p||] = 1$ by $\mathcal{N}(0, 1/(H + 1))$ initialization (Stiennon et al., 2020). We support this through Figure 1, average $||W_p||$ across four seeds for each model after training with $\mathcal{L}_{\text{BT}}$ actually stays near 1.

**Norm variance inflates in hidden states with $\mathcal{L}_{\text{BT}}$**  We continue by analyzing the impact of $||h(x, y)||$ in over-optimization. $\min_\theta \mathcal{L}_{\text{BT}}$ can be achieved through maximizing $\Delta r$, the inner term of sigmoid function:

$$\Delta r_\theta = ||W_p|| \cdot ||h_\theta(x, y_w) - h_\theta(x, y_l)|| \cdot \cos\psi_\Delta, \quad (23)$$

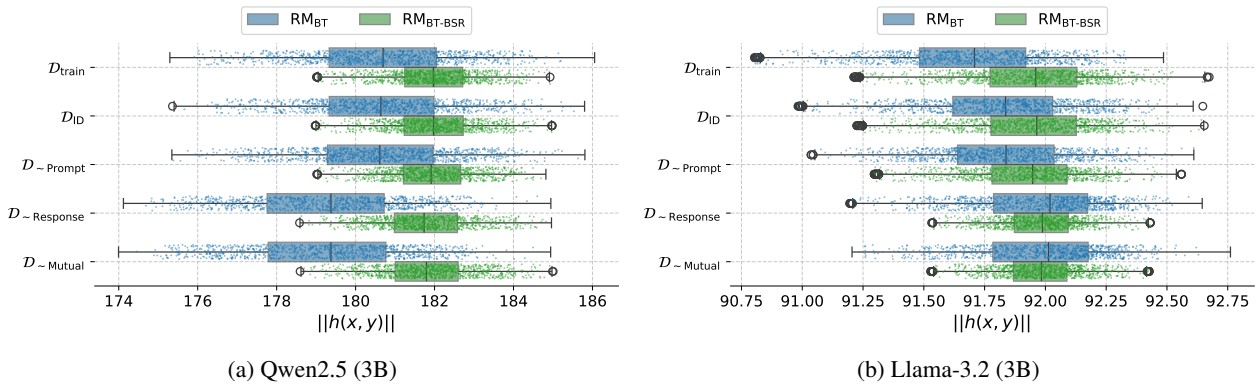

(a) Qwen2.5 (3B)             (b) Llama-3.2 (3B)

*Figure 3.* Hidden state norm dispersion comparison between RM$_{\text{BT}}$ and RM$_{\text{BT-BSR}}$. Batch-wise sum-to-zero regularization (**BSR**) alleviates hidden state norm dispersion and demonstrates a consistent range of norms across different generalization scenarios in Section 2.2. Reducing the variability of hidden state norms, $\mathcal{L}_{\text{BSR}}$ improves the robustness of RMs for unseen data.

when $\cos\psi_\Delta$ is the cosine similarity between $W_p$ and $h_\theta(x, y_w) - h_\theta(x, y_l)$. Having $||W_p||$ stays near its initial value of one, $\mathcal{L}_{\text{BT}}$ encourages $\theta$ to increase $||h_\theta(x, y_w) - h_\theta(x, y_l)||$ and $\cos\psi_\Delta$.

**Over-confidence and over-optimization in RMs** In a multi-class classification task, Wei et al. (2022) points out that excessive growth of logit magnitude in classifiers causes overfitting in the training set: *i.e.*, over-confidence issue. This is closely connected to the over-optimization problem as the reward models are simply classifiers optimized with two-class classification objective $\mathcal{L}_{\text{BT}}$. Viewing $r_\theta(x, y)$ (16) as a logit in a classifier, our analysis narrows the cause of growth in $||r_\theta(x, y)||$ to the growth of $||h_\theta(x, y_w) - h_\theta(x, y_l)||$ as $||W_p||$ remains near one.

In Figure 2, we track the growth of $||h_\theta(x, y_w) - h_\theta(x, y_l)||$ while fine-tuning RM with $\mathcal{L}_{\text{BT}}$. Compared to that of hidden states from the SFT model (Epoch 0), we observe a consistent increase in the average norm and its variance, especially strengthening the right-skewness. This aligns with how the over-confidence issue happens due to growing logit magnitude in typical classification tasks (Wei et al., 2022).

### 4.2. Method: Batch sum-to-zero regularization (BSR)

From this context, we propose regularizing the reward sum of batch to zero to control the inflation in hidden state norm variance, namely *batch sum-to-zero regularization* (**BSR**):

$$\mathcal{L}_{\text{BT-BSR}} = \mathcal{L}_{\text{BT}} + \lambda \cdot \mathcal{L}_{\text{BSR}} \qquad (24)$$

$$\mathcal{L}_{\text{BSR}} = \left( \frac{1}{2|\mathcal{B}|} \sum_{i=1}^{|\mathcal{B}|} \sum_{j \in \{w,l\}} r(x_i, y_{i,j}) \right)^2, \qquad (25)$$

where $\lambda$ is the weight hyperparameter and $\mathcal{B}$ refers to the batch of $(x, y_w, y_l)$. $\mathcal{L}_{\text{BSR}}$ penalizes $r(x, y)$ from being

skewed by enforcing the reward sum to zero, constraining the norm dispersion and outliers in Figure 2.

In detail, the gradient for $h(x, y_w)$ and $h(x, y_l)$ are symmetric, having same magnitude but opposite directions:

$$\frac{\partial \mathcal{L}_{\text{BT}}}{\partial h(x, y_w)} = -\sigma(-\Delta r)W_p \qquad (26)$$

$$\frac{\partial \mathcal{L}_{\text{BT}}}{\partial h(x, y_l)} = \sigma(-\Delta r)W_p. \qquad (27)$$

By $\mathcal{L}_{\text{BT}}$, $||h(x, y_w) - h(x, y_l)||$ incrementally increases as Figure 2. Meantime, $\mathcal{L}_{\text{BSR}}$ penalizes outliers with large magnitude toward either positive or negative directions, shown through its gradient:

$$\frac{\partial \mathcal{L}_{\text{BSR}}}{\partial h(x_i, y_{i,j})} = \left( \frac{1}{|\mathcal{B}|} r(x_i, y_{i,j}) \right) \cdot W_p, \qquad (28)$$

for a prompt and response pair $(x_i, y_{i,j})$ with $i \in \{1, \ldots, |\mathcal{B}|\}$ and $j \in \{w, l\}$, preventing excessive growth as the gradient is proportional to $r(x, y)$. This is a straightforward solution for norm variance inflation in Figure 2, as $r(x, y)$ and its dispersion are largely influenced by $||h(x, y)||$ as discussed in Section 4.1, especially as $r_\theta(x, y_w)$ and $r_\theta(x, y_l)$ are positive and negative values.

We support this point through Figure 3, RM trained with $\mathcal{L}_{\text{BT-BSR}}$ having significantly lower dispersion in $||h(x, y)||$. Comparing the variability of $||h(x, y)||$ on four different generalization scenarios in Section 2.2, reward model trained with $\mathcal{L}_{\text{BT-BSR}}$ (RM$_{\text{BT-BSR}}$) is robust to unseen prompt and responses, while the norm range differs with reward model trained with $\mathcal{L}_{\text{BT}}$ (RM$_{\text{BT}}$).

### 4.3. Baselines

Based on Section 4.1, we adopt three algorithmic mitigations as baseline methods to fulfill Equation (23).

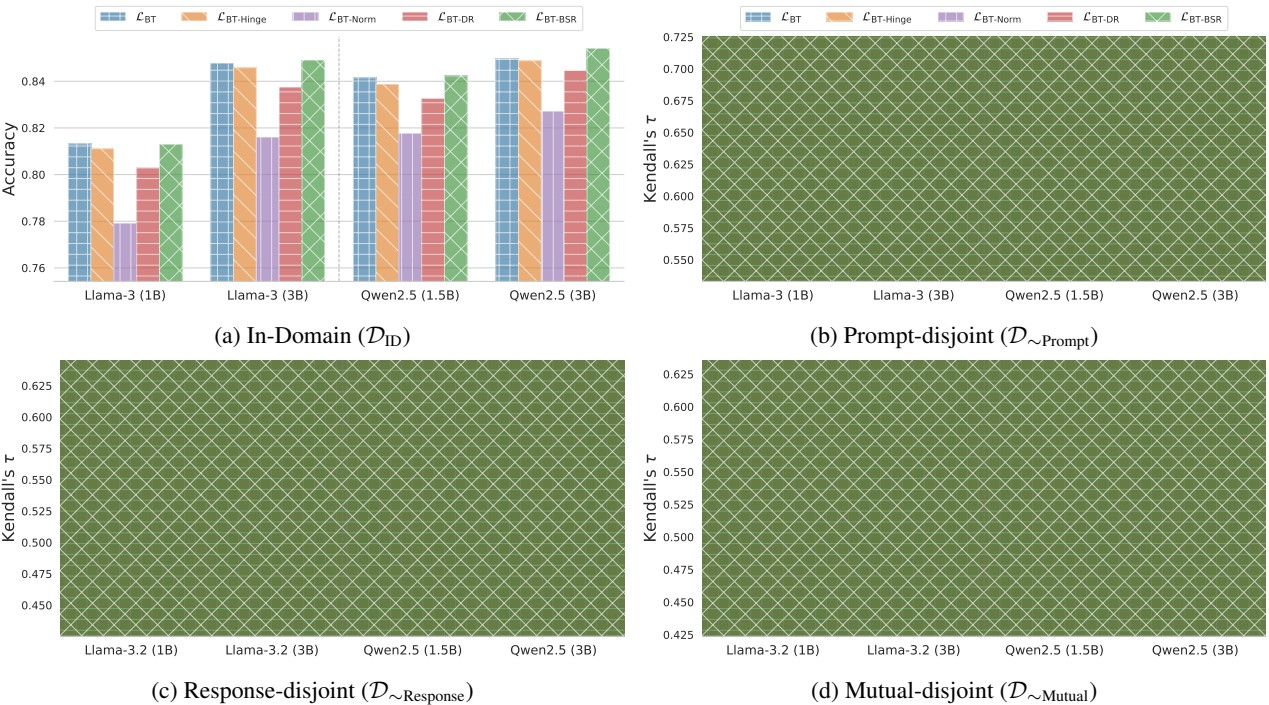

**Figure 4.** Assessing the robustness of each reward modeling objective on four generalization scenarios in Section 2.2. By applying $\mathcal{L}_{\text{BT-BSR}}$, downstream RMs are more robust to unseen prompt sets or response generation model sets by being more aligned to the gold preference model $r^*$, measured through accuracy and Kendall's $\tau$ against the preference annotations of $r^*$. Full results are reported in Table 5.

First, we can explicitly normalize rewards with their norm. Logit normalization (Wei et al., 2022) can be applied to $\mathcal{L}_{\text{BT}}$ as it isolates classification objective to the directions of logits, $\cos\psi$, without being affected by $||h(x,y)||$ by normalizing with the norm of rewards:

$$\mathcal{L}_{\text{BT-Norm}} = -\log\sigma\left(\frac{r(x,y_w) - r(x,y_l)}{\sqrt{r(x,y_w)^2 + r(x,y_l)^2}}\right). \quad (29)$$

Second, we can set the hardbound of $\Delta r$ to explicitly prevent the divergence in Equation (23). We add the hinge loss (Cortes, 1995) that provides hardbound $m$:

$$\mathcal{L}_{\text{BT-Hinge}} = \max\left(0, m - (r(x,y_w) - r(x,y_l))\right). \quad (30)$$

Third, we test the extreme case of boosting the margin in Equation (23), which is expected to reinforce the over-confidence issue. We find the loss function in Yuan et al. (2024) additionally boosting the margin with separate logistic losses for chosen and rejected responses, respectively:

$$\mathcal{L}_{\text{BT-DR}} = \mathcal{L}_{\text{BT}} - \log\sigma(r_\theta(x,y_w)) - \log\sigma(-r_\theta(x,y_l)). \quad (31)$$

Based on our analysis in Section 4.1, we hypothesize that $\mathcal{L}_{\text{BT-DR}}$ will underperform compared to $\mathcal{L}_{\text{BT}}$ in all generalization scenarios by booting the hidden state norm dispersion.

# 5. Results and Analysis

Through three steps, we analyze the significance of robustness in reward models (RMs) in reinforcement learning with human feedback (RLHF). Beginning with how each method affects robustness in RMs through over-optimization assessment in Section 5.1, we study how the robustness in RMs as proxies can boost the alignment toward the true preference in Section 5.2. Finally, we expand our scope to state-of-the-art data and models in Section 5.3.

## 5.1. Part 1: Reward model over-optimization

In Figure 4, we assess the robustness of reward models (RMs) trained with each method in Section 4.3 on four generalization scenarios discussed in Section 2.2. The full results are reported in Appendix C.

**RMs are prone to unseen response styles**  Comparing Figures 4b and 4d, most of the RMs were prone to unseen response styles. While both $\mathcal{D}_{\sim\text{Prompt}}$ and $\mathcal{D}_{\sim\text{Mutual}}$ shares the same prompt set *unseen* during the training, adding an unseen response model set $\mathcal{M}_{\text{valid}}$ triggers more than 10% loss in Kendall's $\tau$ values. For instance, RM$_{\text{BT}}$ experienced 16.4% decrease, from $\tau$ of 0.705 to 0.587. This implies that harnessing diverse LLMs in synthetic preference datasets is crucial for the general use of RMs.

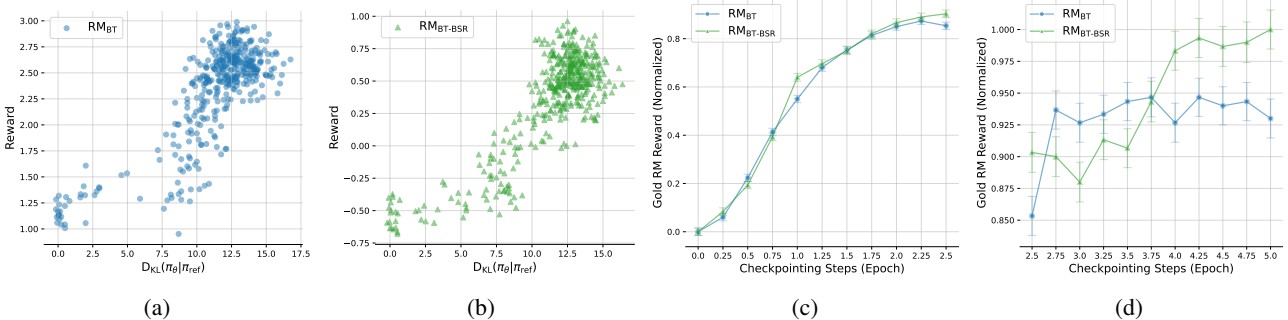

*Figure 5.* Reward to KL divergence plots (Figures 5a and 5b) and alignment to gold reward model $r^*$ (Figures 5c and 5d) while fine-tuning Qwen2.5-1.5B SFT model as $\pi_\theta$ with RLOO using RM$_{BT}$ and RM$_{BT-BSR}$, respectively. We measure the alignment with the gold preference model $r^*$ to verify if reward maximization with each RM as a proxy is leading to maximizing $r^*$ scores in Figures 5c and 5d. Using RM$_{BT-BSR}$ consistently improved the reward assessed by $r^*$, while RM$_{BT}$ stagnated in the last half of the training.

**BSR enhances the robustness of RMs** In Figure 4, RM$_{BT-BSR}$ best aligns to $r^*$ across all generalization scenarios. Meantime, the accuracy of RM$_{BT-DR}$, which is expected to have even stronger norm dispersion, is consistently lower than RM$_{BT}$, evidently supporting our hypothesis. With Figure 3, we inspect that $||h(x,y)||$ remaining stable with unseen data is a crucial condition for robustness in RMs.

Also, RM$_{BT-Hinge}$ occasionally exceeded RM$_{BT}$ in Figures 4c and 4d. Recall that $\mathcal{L}_{BT-Hinge}$ sets hardbound to restrict the divergence in Equation (23); such a result supports that the growth of $||h(x,y)||$ is a major cause of over-optimization as discussed. However, explicit logit normalization with $\mathcal{L}_{BT-Norm}$ underperformed in every scenario. This implies that $||h(x,y)||$ shouldn't be excluded in reward prediction, highlighting the necessity of soft regularization.

Finally, we observe that the performance gap between RM$_{BT}$ and RM$_{BT-BSR}$ is amplified as the model size increases in all four plots, comparing within the same model families. We further study this in Section 5.3 with Llama-3.1-8B.

### 5.2. Part 2: Propagation of RM robustness in RLHF

We study how robust RMs improve RLHF tasks by using RM$_{BT}$ and RM$_{BT-BSR}$ for RLOO training in Figure 5. We refer to $\pi_{BT}$ and $\pi_{BT-BSR}$ to be the two RLOO-trained policies with Qwen2.5-3B based RM$_{BT}$ and RM$_{BT-BSR}$.

**Robustness in RMs lead to better alignment of $\pi$ with $r^*$** In Figures 5c and 5d, we observe $\pi_{BT-BSR}$ being better aligned to the gold preference model $r^*$ (*i.e.*, ArmoRM). While $\pi_{BT}$ and $\pi_{BT-BSR}$ both incrementally maximized the reward in terms of RM$_{BT}$ and RM$_{BT-BSR}$, Figure 5d depicts that the $r^*$ scores *stagnate* in the last half for RLOO training for $\pi_{BT}$. This could imply RM$_{BT}$ providing an imperfect reward signal in the high-reward region, where $||h(x,y)||$ tends to be relatively large. This aligns with the observation in Figure 3 where the average $||h(x,y)||$ of RM$_{BT}$ was

*Table 1.* Effective rank analysis for RM$_{BT}$ and RM$_{BT-BSR}$ by collecting the hidden states of each model. While RM$_{BT}$ structures the train data into a lower rank, the effective rank significantly increases on the unseen evaluation dataset from RM-Bench.

| | **erank**$_{train}$ | **erank**$_{eval}$ | $\Delta\ (\downarrow)$ |
|---|---|---|---|
| RM$_{BT}$ | 23.34 | 33.76 | +10.42 |
| RM$_{BT-BSR}$ | 33.45 | 33.59 | **+0.14** |

sensitive to the unseen data.

**BSR and stable reward maximization** Through Figures 5a and 5b, we compare the stability in reward maximization by tracking the divergence of $\pi_\theta$. With RM$_{BT}$, initial reward maximization required a leap in KL divergence, as shown in the 0.0 to 7.5 range in Figure 5a. On the other hand, RM$_{BT-BSR}$ induced continuous exponential growth in reward while $\pi_{BT-BSR}$ diverged. Along with steady increase in $r^*$ score, this implies that RM$_{BT-BSR}$ yield stable training.

### 5.3. Part 3: Real-world impact of robustness in RM

We apply our method and insights to the state-of-the-art preference dataset and the most widely practiced base model, Skywork-Preferences-v0.2 (SKP) and Llama-3.1-8B, where we do not have access to the true preference model. For RM$_{BT}$, we use the official checkpoint of Liu et al. (2024a)[2].

**Parametric robustness analysis for RM$_{BT}$ and RM$_{BT-BSR}$** Along with RM-bench, we employ the effective rank (Roy & Vetterli, 2007; Wei et al., 2024, erank) as an parametric analysis of robustness. The large discrepancy between the erank of the train and evaluation sets indicates that the model fails to capture information that does not exist in the train

---

[2]https://huggingface.co/Skywork/
Skywork-Reward-Llama-3.1-8B-v0.2

*Table 2.* RM-Bench comparison between RMs trained with $RM_{BT}$ and $BT_{BT-BSR}$ with varying $\lambda \in \{10^{-2}, 10^{-3}, 10^{-4}\}$. By setting $\lambda = 0$, $BT_{BT-BSR}$ is equivalent to $RM_{BT}$. As $\lambda$ gets larger, accuracy in hard tasks with subtle differences increases, consistently exceeding $RM_{BT}$.

| RM-Bench | Chat | Math | Code | Safety | Hard Acc | Normal Acc | Easy Acc | Overall |
|---|---|---|---|---|---|---|---|---|
| $RM_{BT}$ ($\lambda = 0$) | 69.2 | **62.1** | 53.4 | **95.9** | 47.8 | 74.0 | **88.4** | 70.1 |
| $RM_{BT-BSR}$ ($\lambda = 10^{-4}$) | 70.0 | 61.4 | 52.9 | 95.1 | 47.9 | **74.3** | 87.4 | 69.9 |
| $RM_{BT-BSR}$ ($\lambda = 10^{-3}$) | **70.1** | 61.2 | **53.9** | 95.4 | 52.1 | 73.5 | 85.0 | **70.2** |
| $RM_{BT-BSR}$ ($\lambda = 10^{-2}$) | 70.0 | 60.1 | 53.1 | 96.0 | **55.7** | 72.5 | 81.9 | 70.0 |

set but in the general corpus: *i.e.*, overfitted to the train set.

Given a dataset $\mathcal{D}$ with $N$ rows, we collect a set of hidden states $H \in \mathbb{R}^{N \times H}$. To compute the effective rank of $H$, let $Q = \min(N, H)$, and denote the singular values of $H$ by $\sigma_1, \sigma_2, \ldots, \sigma_Q$, we have:

$$\mathbf{erank}(H) = \exp\left( -\sum_{j=1}^{Q} \frac{\sigma_j}{\sum_{i=1}^{Q} \sigma_i} \log \frac{\sigma_j}{\sum_{i=1}^{Q} \sigma_i} \right). \tag{32}$$

In Table 1, we compute the erank of $RM_{BT}$ and $RM_{BT-BSR}$ by collecting their hidden states for SKP ($\mathbf{erank}_{train}$) and RM-Bench ($\mathbf{erank}_{eval}$). While $RM_{BT-BSR}$ retains $\mathbf{erank}_{train}$ in the evaluation set, $RM_{BT-BSR}$ experiences around 40% increase. Thus, $RM_{BT}$ is prone to over-optimization, as previous analyses show. Furthermore, this makes RMs even more vulnerable to dataset biases, such as verbosity bias (Saito et al., 2023; Zhang et al., 2024b; Chen et al., 2024a).

**BSR captures subtle preference factors**    When $r^*$ is not available, we evaluate RMs through RM-Bench (Liu et al., 2025b), which provides preference pairs with subtle differences and achieves higher correlation against actual use cases than RewardBench (Lambert et al., 2025).

In Table 2, we observe RM's accuracy consistently increasing in the response pairs with subtle differences ("Hard Acc") as $\lambda$ gets larger. Liu et al. (2025b) reports that *hard accuracy* of RM in RM-Bench has the highest correlation with the downstream policy's actual performance, while the correlation was near zero for *easy accuracy*. Along with the fact that the over-confidence issue hinders accurate classification in hard tasks (Wei et al., 2022), we inspect that $RM_{BT}$ fails to capture subtle differences in their representations, also supported by a significant increase in erank for unseen data in Table 1.

For final RLHF training, we select $RM_{BT-BSR}$ with $\lambda = 10^{-3}$ in Table 2, regarding a balance between the hard and easy tasks along with the overall scores. We report the reward logs during the training for both in Appendix D.

**$RM_{BT-BSR}$ boosts AlpacaEval without verboseness**    We evaluate $\pi_{BT}$ and $\pi_{BT-BSR}$ through length-controlled (LC) Alpacaeval 2.0 (Dubois et al., 2024), which GPT-4 (OpenAI

*Table 3.* AlpacaEval 2.0 average response length and length controlled win ate for the checkpoints throughout the RLHF process using Qwen2.5-1.5B with SFT on TULU3 mixture. Using $RM_{BT+BSR}$ significantly reduces verboseness while being best aligned.

| Qwen2.5-1.5B | Length | LC AE2.0 (%) |
|---|---|---|
| SFT | 2247 | 2.59 |
| + RLOO ($RM_{BT}$) | 2180 | 8.41 |
| + RLOO ($RM_{BT+BSR}$) | **1337** | **9.02** |

et al., 2024) is used as LLM-as-a-judge. In Table 3, $\pi_{BT-BSR}$ demonstrates the highest LC win rate with a significantly shorter response than the SFT model and $\pi_{BT}$.

As preferring verbose response is a chronic problem in generative evaluators (Zheng et al., 2023; Dubois et al., 2024), reducing generation length by 40% compared to SFT model while achieving the best win rate in Table 3 highlights how $RM_{BT-BSR}$ could be advantageous for preventing policies falling into local minima while maximizing the reward: *i.e.*, reward gaming (Skalse et al., 2022; Pang et al., 2023; Chen et al., 2024b). Comprehensively, results underscore the effectiveness of $\mathcal{L}_{BT-BSR}$ for the overall RLHF pipeline.

## 6. Related Works

**Over-optimization of reward models**    As formally outlined by Gao et al., 2023, *reward over-optimization* refers to the phenomenon where the reward model (RM) fails to generalize to the gold objective due to excessive training. To enhance out-of-distribution generalizability some works have proposed ensembling RMs (Gleave & Irving, 2022; Zhai et al., 2023; Coste et al., 2024), meta-learning RMs on the shifted target distribution (Wang et al., 2024a), augmenting preference data for RMs (Liu et al., 2025a), jointly learning the text-generation loss in reward modeling (Yang et al., 2024b), or constraining the preference optimization process itself (Moskovitz et al., 2024; Zhang et al., 2024a; Liu et al., 2024b; Gupta et al., 2025). Interestingly, Rafailov et al., 2024 outlined how a similar phenomenon can be seen in Direct Alignment Algorithms (DAAs), where the implicit rewards of the policy replace RM. On the other hand, some

applications on multiliguality (Wu et al., 2024; Hong et al., 2025) have suggested how RMs can possess cross-lingual transfer to languages unseen during reward modeling.

**Evaluating reward models** Parting from assessing the accuracy of the validation sets of popular preference datasets (Stiennon et al., 2020; Bai et al., 2022), Lambert et al., 2025 formalized a reproducible toolkit to evaluate reward models for enhanced explainability in diverse preference domains and tasks. Subsequent works have addressed improvements in cases of handling more sensitive, subtle cases (Liu et al., 2025b) and in wider coverage of real-world scenarios (Zhou et al., 2025) or specializations in tasks such as mathematical reasoning (Kim et al., 2024). Frick et al., 2024 introduced an evaluation pipeline directly targeted at assessing the RM in its role in the reinforcement learning with human feedback (RLHF) pipeline.

## Conclusion

This study outlines why reward models (RMs) trained with the Bradley-Terry (BT) model loss can be vulnerable to over-optimization issues, losing generalizability to unseen tasks. We highlight the dispersion in the norm of hidden states in RMs as a primary source of such issue with theoretical analysis and empirical demonstrations across model families and sizes. Then, we propose batch-wise sum-to-zero regularization (**BSR**), an add-on to the BT model that penalizes the rewards for having an abnormally large magnitude. We present threefold experiments throughout the overall RLHF pipeline, starting by assessing the robustness of BSR and four baseline methods through the alignment against the synthetic gold preference model. Then, we RLHF training with RLOO using RMs trained with BT model and BSR, respectively. We observe a stronger alignment of the resulting policy after RLOO against the gold preference model. Eventually, we expand the experiments in 8B size model with high-quality preference data, where RM with BSR surpasses the state-of-the-art RM in 8B size with 7% improvements.

## Impact Statement

This paper aims to identify the causes of reward model over-optimization and suggests a simple method to enhance robustness in reward models. As reward models are proxies for human preferences in language model alignment, the proposed method has the potential to impact our society, none of which we feel must be specifically highlighted here.

## Acknowledgement

This work was supported by Institute for Information & communications Technology Planning & Evaluation(IITP) grant funded by the Korea government(MSIT) (RS-2024-00398115, Technology research to ensure authenticity and consistency of results generated by AI) and (RS-2019-II190075, Artificial Intelligence Graduate School Program (KAIST)).

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

# A. Training Configurations

For supervised fine-tuning (SFT) and reward modeling, we use Liger-Kernel (Hsu et al., 2024) with DeepSpeed ZeRO-3 (Rajbhandari et al., 2020) and FSDP (Zhao et al., 2023) for efficient training. Including reinforcement learning with human feedback (RLHF) phase, we utilize the TRL library (von Werra et al., 2020) to adjust to our usage. We used NVIDIA A100 and A6000 GPUs throughout the experiments.

## A.1. Supervised fine-tuning

We train every model on UltraChat for a single epoch with a global batch size of 512 following Tunstall et al. (2024). We set a learning rate of $10^{-5}$ with 10% warmup and cosine decay.

## A.2. Reward modeling

We train reward models on top of the SFT models above with four different seeds. We fix the global batch size of 128 across the models and methods. We set a learning rate of $5 \times 10^{-6}$ for Llama-3.2-1B and Qwen2.5-1.5B models, $3 \times 10^{-6}$ for 3B models, and $2 \times 10^{-6}$ for Llama-3.1-8B and Qwen2.5-7B models. 5% warmup and linear decay were applied following Lambert et al. (2024). We use FSDP for distributed training in reward modeling. We set $\lambda = 10^{-3}$ in Section 5.1 and ablate different $\lambda$ in Section 5.3.

## A.3. Reinforcement learning with human feedback

We leverage Async RLHF (Noukhovitch et al., 2025) with vLLM (Kwon et al., 2023) as a backend to reduce the bottleneck in on-policy generations. To test reward models with varying sizes under controlled training configurations (*e.g.*, batch size), we separately deploy the reward models following OpenRLHF (Hu et al., 2024). We use a batch size of 1 for every reward model to prevent potential distortions in reward scores with padding applied.

*Table 4.* RLOO training configuration details for each section. We train Qwen2.5-1.5B with SFT using corresponding reward models for each section using 4 A6000 GPUs, excluding the GPUs assigned for the reward model and vLLM engine for on-policy generation.

| Category | Section 3.2 | Section 3.3 |
|---|---|---|
| **Learning Rate** | $2 \times 10^{-6}$ | $1 \times 10^{-6}$ |
| $\beta$ | 0.05 | 0.05 |
| **Number of responses ($k$)** | 2 | 2 |
| **Global Batch (Effective)** | 128 | 128 |
| **Learning Rate Scheduler** | Linear Decay | Linear Decay |
| **Warmup Ratio** | 0.03 | 0.03 |
| **Training Epochs** | 5 | 5 |

# B. Data Preprocessing

We adopt UltraFeedback[3] (Cui et al., 2025) for our experiments[4]. Before splitting the train and validation sets, we filtered items with duplicated prompts and responses. We first removed the items with identical prompts and removed the rows when at least two responses out of four were identical. As a result, we used 62,282 rows after filtration from 63,966.

In detail, 17 models comprising $\mathcal{M}_{\text{train}}$ for UF are: GPT-3.5-Turbo , GPT-4 (OpenAI et al., 2024), Bard[5], Llama-2-7B-Chat, Llama-2-13B-Chat, Llama-2-70B-Chat (Touvron et al., 2023), WizardLM-7B, WizardLM-13B, WizardLM-70B (Xu et al., 2024), Vicuna-33B (Chiang et al., 2023), Alpaca-7B (Taori et al., 2023), Falcon-40B-Instruct (Almazrouei et al., 2023), MPT-30B-Chat (Team, 2023), StarChat-beta (Tunstall et al., 2023), and Pythia-12B (Biderman et al., 2023).

Since large proportion of the models in $\mathcal{M}_{\text{train}}$ are Llama-2 based models, we selected distinct model families for $\mathcal{M}_{\text{valid}}$, including OLMO2-7B-Instruct (OLMo et al., 2025), SmolLM2-1.7B-Instruct (Allal et al., 2025), Mistral-Instruct-v0.2 (Jiang et al., 2023a), and Gemma-2-2B-It (Team et al., 2024).

**Train set** ($\mathcal{D}_{\text{train}}$)   First, we select a random set of 51,200 samples from UF as the *train set*. Then, we choose two random responses out of four for each prompt in the train set. Thereby, we have 51,200 triplets comprising prompt and corresponding chosen and rejected responses, according to ArmoRM.

**Validation 1 - in-domain** ($\mathcal{D}_{\text{ID}}$)   Then, we use the remaining two responses in 51,200 prompts in the train set as the *in-domain validation set* to evaluate if the trained reward models can generalize in **same prompt and response spaces**. Since we have two responses per prompt, we use binary accuracy as an evaluation metric.

**Validation 2 - prompt out-of-domain** ($\mathcal{D}_{\text{Prompt-OOD}}$)   We set the remaining 12,800 instances as a *prompt out-of-domain (Prompt OOD)* validation set, having **different prompt space but same response space**. Since we have four responses per prompt, we use Kendall's $\tau$ ranking correlation (Kendall, 1962) as an evaluation metric.

**Validation 3 - response out-of-domain** ($\mathcal{D}_{\text{Response-OOD}}$)   For the prompts in the train set, we additionally generate four different responses from the new models: Gemma-2-2B-It (Team et al., 2024), Olmo2-7B-Instruct[6](OLMo et al., 2025), SmolLM2-1.7B-Instruct (Allal et al., 2025), and Mistral-Instruct-v0.2 (Jiang et al., 2023a). By having *response out-of-domain (Response OOD)* validation set, we assess reward models when **prompt space stays the same, but the response space differs**. Since we have four responses per prompt, we use Kendall's $\tau$ as an evaluation metric.

**Validation 4 - mutual out-of-domain** ($\mathcal{D}_{\text{Mutual-OOD}}$)   Using the same models in the Response OOD set, we generate the responses for the prompts from the Prompt OOD set, having *mutual out-of-domain (Mutual OOD)* validation set. Here, we test the model's robustness when **both the prompt and response spaces are distinct**. Since we have four responses per prompt, we use Kendall's $\tau$ as an evaluation metric.

---

[3]https://huggingface.co/datasets/openbmb/UltraFeedback
[4]The final dataset can be found in: https://huggingface.co/rm-robustness
[5]https://blog.google/technology/ai/bard-google-ai-search-updates/
[6]https://huggingface.co/allenai/OLMo-2-1124-7B-Instruct

## C. Robustness Assessment with Gold Preference Model

We report the full results of visualizing Figure 5 in Table 5.

*Table 5.* Preference prediction accuracy over different types of validation sets.

| Model | Method | In-Domain (Accuracy ↑) | Prompt OOD (Kendall's $\tau$ ↑) | Response OOD (Kendall's $\tau$ ↑) | Mutual OOD (Kendall's $\tau$ ↑) |
|---|---|---|---|---|---|
| Llama-3.2 (1B) | $\mathcal{L}_{BT}$ (Bradley & Terry, 1952) | **0.8132** | 0.6157 | 0.5115 | 0.5065 |
| | $\mathcal{L}_{BT\text{-}Hinge}$ (Cortes, 1995) | 0.8111 | 0.6102 | 0.5168 | 0.5131 |
| | $\mathcal{L}_{BT\text{-}Norm}$ (Wei et al., 2022) | 0.7792 | 0.5580 | 0.4503 | 0.4488 |
| | $\mathcal{L}_{BT\text{-}DR}$ (Yuan et al., 2024) | 0.8029 | 0.6015 | 0.493 | 0.4937 |
| | $\mathcal{L}_{BT\text{-}BSR}$ (*Ours*) | 0.813 | **0.6198** | **0.5246** | **0.5206** |
| Qwen2.5 (1.5B) | $\mathcal{L}_{BT}$ (Bradley & Terry, 1952) | 0.8415 | 0.6835 | 0.5601 | 0.5566 |
| | $\mathcal{L}_{BT\text{-}Hinge}$ (Cortes, 1995) | 0.8388 | 0.6756 | 0.5592 | 0.5561 |
| | $\mathcal{L}_{BT\text{-}Norm}$ (Wei et al., 2022) | 0.8177 | 0.6403 | 0.4502 | 0.4507 |
| | $\mathcal{L}_{BT\text{-}DR}$ (Yuan et al., 2024) | 0.8326 | 0.6625 | 0.5187 | 0.5201 |
| | $\mathcal{L}_{BT\text{-}BSR}$ (*Ours*) | **0.8423** | **0.6849** | **0.5618** | **0.5573** |
| Llama-3.2 (3B) | $\mathcal{L}_{BT}$ (Bradley & Terry, 1952) | 0.8478 | 0.679 | 0.6195 | 0.609 |
| | $\mathcal{L}_{BT\text{-}Hinge}$ (Cortes, 1995) | 0.8458 | 0.6829 | 0.6210 | 0.61 |
| | $\mathcal{L}_{BT\text{-}Norm}$ (Wei et al., 2022) | 0.8161 | 0.6272 | 0.5542 | 0.5509 |
| | $\mathcal{L}_{BT\text{-}DR}$ (Yuan et al., 2024) | 0.8375 | 0.6638 | 0.5916 | 0.5849 |
| | $\mathcal{L}_{BT\text{-}BSR}$ (*Ours*) | **0.8491** | **0.702** | **0.6402** | **0.6306** |
| Qwen2.5 (3B) | $\mathcal{L}_{BT}$ (Bradley & Terry, 1952) | 0.8496 | 0.705 | 0.5917 | 0.587 |
| | $\mathcal{L}_{BT\text{-}Hinge}$ (Cortes, 1995) | 0.8488 | 0.6984 | 0.5784 | 0.5685 |
| | $\mathcal{L}_{BT\text{-}Norm}$ (Wei et al., 2022) | 0.8272 | 0.6605 | 0.5295 | 0.5282 |
| | $\mathcal{L}_{BT\text{-}DR}$ (Yuan et al., 2024) | 0.8446 | 0.6855 | 0.5602 | 0.5564 |
| | $\mathcal{L}_{BT\text{-}BSR}$ (*Ours*) | **0.8541** | **0.7202** | **0.5991** | **0.6106** |

## D. Training logs for Section 5.3

We report additional training logs for RLOO training with Skywork-Reward-Llama-3.1-8B-v0.2 ($RM_{BT}$) and $RM_{BT\text{-}BSR}(\lambda = 10^{-3})$ based on Llama-3.1-8B in Table 2.

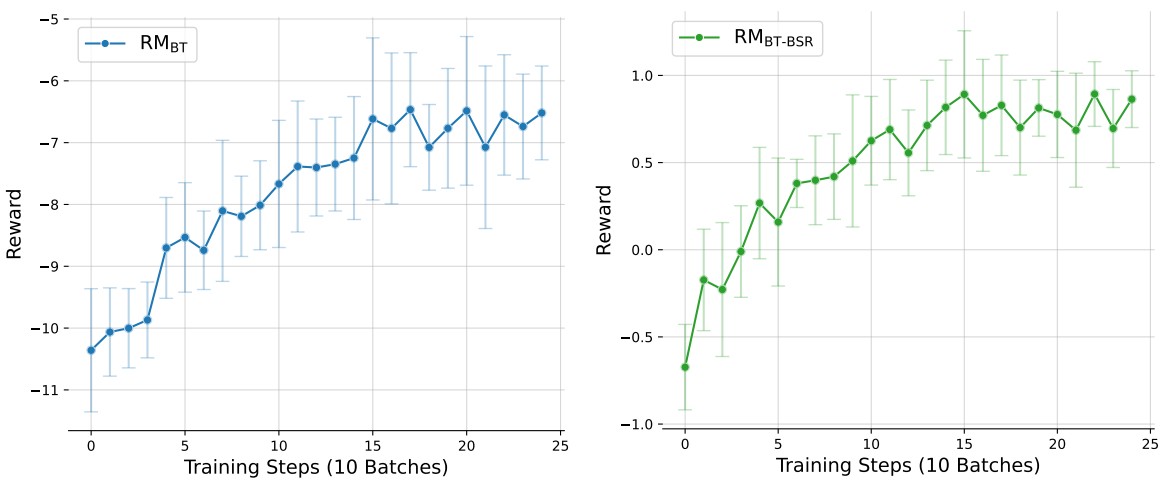

(a) RLOO with Skywork-Reward-Llama-3.1-8B-v0.2 ($RM_{BT}$)     (b) RLOO with Llama-3.1-8B $RM_{BT\text{-}BSR}(\lambda = 10^{-3})$

*Figure 6.* Reward maximization trajectories with Skywork-Reward-Llama-3.1-8B-v0.2 ($RM_{BT}$) and Llama-3.1-8B $RM_{BT\text{-}BSR}$.

