# OpenReview forum: "On the Robustness of Reward Models for Language Model Alignment"
_ICML.cc/2025/Conference — ICML 2025 poster_

### Official Review · Reviewer_YT72 · 2025-03-04

**Overall Recommendation:** 3

**Summary:**

The paper provides a new theoretical analysis framework to understand the robustness of RM in LLMs.

**Claims And Evidence:**

Yes, I think most of the claims made in the submission clear and convincing.

**Essential References Not Discussed:**

Potentially related works missing:
1. Yan Y, Lou X, Li J, et al. Reward-robust rlhf in llms[J]. arXiv preprint arXiv:2409.15360, 2024.

**Experimental Designs Or Analyses:**

Most of the experimental designs are sound and valid. However, there are small points that need to be clarified. See Questions For Authors.

**Methods And Evaluation Criteria:**

See Questions For Authors part.

**Other Comments Or Suggestions:**

1.  line 909: ]

2.  Section 3 should be named as "Experiment Settings" as no results are shown in this section.

**Other Strengths And Weaknesses:**

See Questions For Authors.

**Questions For Authors:**

1.  Is there any quantification method to quantify the disjointness between each datasets mentioned in Section3?&#x20;

2.  Line 184, left column, "Typically, the norm of two vectors ||Wp|| · ||h(x, y)|| largely contributes to maximize the softmax value and cause over-confidence issue, especially in the context of large language models (LLMs) that have large hidden sizes." Can you explain the sentence in detail? What's more, in RM training process, there is no softmax (the language modeling head is replaced by value head), so what is the relationship between this sentence and the context?

3.  Is over-confidence the same as overoptimization?

4.  For the result in Figure 5(c) and (d), why is the change in the two curves discontinuous despite being caused by the same loss? For example, in RM\_BT, during epoch 2.25-2.5, the curve starts to decline, but during epoch 2.5-2.75, the curve again boosts. Why?

**Relation To Broader Scientific Literature:**

The paper provides a valuable vision for the current understanding of RM/RLHF robustness.

**Theoretical Claims:**

I didn't check all the proofs in detail but the theorems provides seem to be reasonable and sounding.

---

> ### Author Rebuttal · Authors · 2025-04-01
>
> We appreciate your suggestions on the typo and the section title. We will make sure to address them in the final version of our paper.
>
> &nbsp;
>
> **Q1 - Quantifying disjointness between datasets:**  We appreciate the reviewer’s question for disjointness quantification as it would make the splitting criteria clearer. We originally planned on splitting the datasets based on the cosine distances of the embeddings of the prompt and responses. However, due to the diverse split settings outlined in Section 2.2, we were unable to provide a strict quantification rule of thumb for disjointness quantifications.
>
> **Q3 - Over-confidence vs over-optimization:**  For clarity, we address the third question before the second one. We view the over-confidence issue, which is originally studied for the conventional multi-class classifiers, as a core cause of the reward model over-optimization problem in the context of reward modeling formulation. [1] shows that an overly growing magnitude of logits triggers classifiers to be over-confident due to the nature of the softmax function. In this paper, we carefully analyze how the backbone model’s hidden state norm is a core cause of the growth in reward magnitude (i.e., which is equivalent to the logit magnitude) as they are a unique type of classifier with two classes but with a single projection head, unlike conventional multi-class classifiers. We continue how over-confidence analysis in RMs connects to over-optimization in Q2.
>
> **Q2 - Clarifications on lines 184-187:** This explanation intends to describe RMs’ formulation as a special case of conventional classifier models, where [1] studies the over-confidence problem, and builds how this over-confidence issue in RM formulation is connected to the over-optimization problem. Reward modeling loss $\mathcal{L}\_\mathrm{BT}$ can be written as a two-class softmax classification. While the over-confidence problem originates from growing logit magnitude in conventional classifiers [1], we are analyzing this problem in the RM context. As the magnitude of the reward (i.e., logit) comes largely from $||h(x, y_||$ by $||W_p|| \simeq 1$ even after training, as we showed, the over-confidence problem in RMs stems from the dispersion of hidden states. As the hidden dimension grows (which is common in larger language models), the norm tends to increase, thereby potentially exacerbating over-optimization issues. This is why these lines of explanation bridge the important connection between the over-confidence and over-optimization issues in RMs.hmm (briefly talked about utilizing embedding model + cosine distance for quantification)
>
> **Q4 - Clarification in Figures 5(c) and 5(d):**  Appendix A.3 explains that we used a linear decay scheduler during RLOO training. Consequently, the later phase of training operates with a lower learning rate, which leads to slower parameter updates and a tendency for the model to overfit the prompt over multiple epochs. Moreover, note that the y-axis scales differ between Figures 5(c) and 5(d): Figure 5(c) covers a range of [0.0, 1.0], while Figure 5(d) focuses on [0.85, 1.0] to allow a more fine-grained comparison between the policies trained with $RM_{\text{BT}}$ and $RM_{\text{BT-BSR}}$. In this context, the result can be interpreted as $RM_{\text{BT}}$ stagnating in the later phase of training, reaching about 95% of the maximum reward, while $RM_{\text{BT-BSR}}$ reaches that level by epoch 5. The transient dip and subsequent boost in the curve likely reflect the combined effects of the decaying learning rate and overfitting dynamics, resulting in non-monotonic updates to the proxy reward signal.
>
>
> &nbsp;
>
> **Reference**
>
> [1] Wei, Hongxin, et al. "Mitigating neural network overconfidence with logit normalization." International conference on machine learning. PMLR, 2022.

---

### Official Review · Reviewer_hrWq · 2025-03-11

**Overall Recommendation:** 3

**Summary:**

This paper investigates the issue of over-optimization in reward models (RMs) within RLHF, identifying excessive hidden state norm dispersion as a key factor. To address this, the authors introduce batch-wise sum-to-zero regularization (BSR), which constrains reward magnitudes by ensuring batch-level zero-centering. They further categorize four generalization scenarios in reward modeling to analyze robustness and over-optimization. Their proposed method outperforms baseline methods and achieves promising experimental results.

**Claims And Evidence:**

Yes

**Essential References Not Discussed:**

None

**Experimental Designs Or Analyses:**

Yes I checked the evaluation on in-domain, prompt-disjoint, response-disjoint and mutual-disjoint cases. Also I checked the experiments that validates the hypothesis that over-optimization is related to the unconstrained growth of the h(x,y). The design seems reasonable.

**Methods And Evaluation Criteria:**

The evaluation settings follow (Gao, 2023) to set the synthetic gold RM instead of real datasets with human annotations. This makes sense to do controlled experiments on reward modeling.

**Other Comments Or Suggestions:**

None

**Other Strengths And Weaknesses:**

Strengths:
1. The paper is very well-written, with each message clearly conveyed in the experimental section.
2. The topic is relevant since reward hacking is a long-standing and unsolved question in the LLM space.
3. The evaluation on in-domain, prompt-disjoint, response-disjoint and mutual-disjoint cases is very useful information.

**Questions For Authors:**

Does the proposed normalization techniques also work for DPO-style methods, where there is no explicit training of a separate reward model?

**Relation To Broader Scientific Literature:**

The paper addresses reward hacking problem by adding proper normalization during training, which is orthogonal to prior work that introduces other loss functions (such as penalty in output length [1]).

[1] Chen, Lichang, et al. "ODIN: Disentangled Reward Mitigates Hacking in RLHF." International Conference on Machine Learning. PMLR, 2024.

**Theoretical Claims:**

No theoretical contents.

---

> ### Author Rebuttal · Authors · 2025-04-01
>
> We thank the reviewer for the insightful comments and will add the suggested paper to the related works.
>
> &nbsp;
>
> **Q1 - Normalization techniques to implicit rewards of DAAs**: DAAs use the language model as an implicit reward model, differing from our classifier-based reward models in that they use a sum of log ratios rather than a linear projection. As our normalization, BT-BSR, starts with the solid background on the over-confidence issue that classifiers are prone to, it may not be intuitive to directly apply batch-wise sum-to-zero constraint on implicit RMs. Applying a batch-wise sum-to-zero constraint directly to implicit rewards, which often exhibit negative scales [1], would be counterintuitive.
>
> Nevertheless, over-optimization can also occur with implicit rewards [2], as evidenced by their divergence to negative infinity [3]. As such, divergence is similar to hidden state norm dispersion in our scenario, the straightforward solution could be Z-loss formulation [4,5]:
>
> $$
> \mathcal{L}\_{IRM-BSR} = - \lambda \log^2 \sum\_i^{N} \left( \beta \log \frac{\pi\_\theta(y\_{i, w}|x)}{\pi\_\mathrm{ref}(y\_{i, w}|x)} + \beta \log \frac{\pi\_\theta(y\_{i, l}|x)}{\pi\_\mathrm{ref}(y\_{i, l}|x)}  \right).
> $$
>
> Z-loss is a well-established method for stabilizing large-scale language model pre-training by penalizing divergence in logits. Here, $\mathcal{L}\_\mathrm{IRM-BSR}$ prevents the implicit reward (log-ratio) from diverging to negative values, mirroring how Z-loss regulates the softmax normalizing constant.
>
> We conducted a minimal experiment on Qwen2.5-1.5B trained with TULU3-SFT data [6] comparing plain DPO with DPO plus the $\mathcal{L}\_\mathrm{IRM-BSR}$ regularizer. As shown in Tables A, B, and C, the Z-loss-based regularizer substantially mitigates negative divergence in log-likelihood while preserving training preference accuracy.
>
> &nbsp;
>
> |        | 0   | 5   | 10  | 15  | 20  | 25  | 30  | 35  | 40  | 45  | 50  | 55  | 60  | 65  | 70  | 75  | 80  | 85  | 90  |
> |--------|-----|-----|-----|-----|-----|-----|-----|-----|-----|-----|-----|-----|-----|-----|-----|-----|-----|-----|-----|
> | DPO    | -302| -242| -274| -306| -378| -322| -314| -334| -460| -434| -448| -426| -528| -414| -440| -376| -474| -466| -424|
> | DPO+Reg| -249| -260| -270| -292| -340| -296| -251| -304| -366| -384| -364| -344| -378| -320| -356| -306| -392| -388| -324|
> > **Table A**: Log-likelihood of chosen responses during training.
>
> |        | 0   | 5   | 10  | 15  | 20  | 25  | 30  | 35  | 40  | 45  | 50  | 55  | 60  | 65  | 70  | 75  | 80  | 85  | 90  |
> |---|---|---|---|-----|-----|-----|-----|-----|-----|-----|-----|-----|-----|-----|-----|-----|-----|-----|-----|
> | DPO    | -340| -296| -366| -430| -492| -432| -484| -476| -488| -612| -572| -632| -740| -596| -652| -564| -656| -604| -620|
> | DPO+Reg| -314| -318| -338| -422| -478| -434| -384| -458| -474| -524| -470| -544| -572| -508| -498| -450| -532| -506| -520|
> > **Table B**: Log-likelihood of rejected responses during training.
>
> |        | 0     | 5     | 10    | 15    | 20    | 25    | 30    | 35    | 40    | 45    | 50    | 55    | 60    | 65    | 70    | 75    | 80    | 85    | 90    |
> |---|---|----|----|---|---|--|---|--|--|--|---|---|-|--|-|---|--|---|-|
> | DPO    | 0.2125 | 0.375  | 0.5125 | 0.625  | 0.6    | 0.625  | 0.675  | 0.675  | 0.6875 | 0.7125 | 0.6875 | 0.7625 | 0.6999 | 0.7125 | 0.75   | 0.6999 | 0.6625 | 0.75   | 0.675  |
> | DPO+Reg| 0.1875 | 0.4    | 0.5375 | 0.6062 | 0.6000 | 0.6687 | 0.6875 | 0.6812 | 0.7437 | 0.6937 | 0.6875 | 0.7312 | 0.8125 | 0.6875 | 0.7437 | 0.6437 | 0.7624 | 0.7812 | 0.7312 |
> > **Table C**: Preference accuracy for each batch during training.
>
> &nbsp;
>
>
>
> &nbsp;
>
>
> **Reference**
>
> [1] Rafailov, Rafael, et al. "From $ r $ to $ Q^* $: Your Language Model is Secretly a Q-Function." First Conference on Language Modeling.
>
> [2] Rafailov, Rafael, et al. "Scaling laws for reward model overoptimization in direct alignment algorithms." Advances in Neural Information Processing Systems 37 (2024): 126207-126242.
>
> [3] Shi, Zhengyan, et al. "Understanding likelihood over-optimisation in direct alignment algorithms." arXiv preprint arXiv:2410.11677 (2024).
>
> [4] Chowdhery, Aakanksha, et al. "Palm: Scaling language modeling with pathways." Journal of Machine Learning Research 24.240 (2023): 1-113.
>
> [5] Wortsman, Mitchell, et al. "Small-scale proxies for large-scale Transformer training instabilities." The Twelfth International Conference on Learning Representations.
>
> [6] Lambert, Nathan, et al. Tulu 3: Pushing frontiers in open language model post-training." arXiv preprint arXiv:2411.15124 (2024).

---

### Official Review · Reviewer_Ajeh · 2025-03-13

**Overall Recommendation:** 1

**Summary:**

This paper explores the challenges of over-optimization in reward models used in RLHF of LLMs. It identifies the dispersion of hidden state norms as a primary cause of over-optimization and proposes batch-wise sum-to-zero regularization (BSR) to address this by penalizing outliers and controlling reward dispersion. The study demonstrates that BSR not only improves the robustness of reward models but also enhances their performance on complex preference prediction tasks across different datasets.

**Claims And Evidence:**

The preliminary theoretical derivation makes sense. However, the claims for different scenarios might need more comprehensive experiments to be robust and convincing.

**Essential References Not Discussed:**

See above.

**Experimental Designs Or Analyses:**

As mentioned above, the experiments are not comprehensive enough and I have concerns about the practical utility of most of the designed evaluation metrics in the discussion.

**Methods And Evaluation Criteria:**

The use of batch-wise sum-to-zero regularization (BSR) in the paper is a sensible approach to addressing over-optimization in reward models by controlling reward dispersion. Nonetheless, the paper’s discussions and experiments lack of focus and the paper is quite hard to read. Various scenarios are considered but for each of them, the experiments are not comprehensive enough (for example consider more models and datasets) to make the results convincing.
Furthermore, it’s important to note that the idea of centering rewards at zero is not entirely novel, as similar concepts have already been explored in existing models (for example see the discussion in [1].

[1] Lambert, N., Pyatkin, V., Morrison, J., Miranda, L. J., Lin, B. Y., Chandu, K., ... & Hajishirzi, H. (2024). Rewardbench: Evaluating reward models for language modeling. arXiv preprint arXiv:2403.13787.

**Other Comments Or Suggestions:**

N/A

**Other Strengths And Weaknesses:**

The exploration of various scenarios and the provision of hypotheses and insights are strengths of the paper.

**Questions For Authors:**

N/A

**Relation To Broader Scientific Literature:**

The key contribution is on the sum-to-zero regularization, but this idea might lack novelty.

**Theoretical Claims:**

There are some preliminary theoretical derivations which looks correct. There are no technical theoretical claims or proofs.

---

> ### Author Rebuttal · Authors · 2025-04-01
>
> We thank the reviewer for the comment and would like to further discuss the addressed points.
>
> &nbsp;
>
> **W1 - Limited Experimental Results**: Our experimental design can be streamlined into threefold: (1) assessing alignment between proxy RMs and gold RMs with different learning objectives, (2) propagation of over-optimization in RMs to their downstream usage for RLHF training, and (3) understanding real-world impact of such propagation with state-of-the-art model and data. This line of analysis is fully supported with Qwen2.5 and Llama3.2 series with 1B and 3B scale on carefully curated UltraFeedback, followed by Llama-3.1-8B model and 80k Skywork-Preferences dataset. While the clarity in writing could be a valid concern if the logical flow is not fully conveyed, we believe the paper encompasses a large range of models and datasets, especially aligning with practical usage in the research community (e.g., the majority of RMs in RewardBench [1] are based on Llama-3.1 or Llama-3.2 series and Skywork-Preferences dataset).
>
> To ensure that the concerns on the width of experiments, we extend the experiment of Figure 5 (Section 5.2) by scaling the model to Qwen2.5-3B in Table A. This result further validates that the trend shown in Qwen2.5-1.5B on the propagation of RM robustness in the RLHF stage on a larger scale. If the concerns about the clarity of writing can be specified, we would happily provide further explanation and an update in the final version of the paper.
>
> &nbsp;
>
>
> **W2 - Novelty of Batch-wise Sum-to-Zero Constraint in Reward Modeling**: We appreciate the concern on methodological novelty, but there seems to be a major confusion on the zero-centering aspect of BSR. While [1] is cited as the core reference to point out that the idea of centering rewards to zero is not novel, we do not find it to be connected to our approach:
>
> > (Excerpt from Discussion section of RewardBench according to reviewer’s pointer)...Few RMs are Gaussian in their scores across the REWARDBENCH datasets, fewer RMs are centered around 0 reward, and none we tested centered Gaussians. Future work should identify a preferred RM output distribution for downstream RL training.
>
> This is the only part that we find in [1] that mentions centering. However, they intend that RMs are rarely centered through post-hoc analysis. To the best of our knowledge, [2] is the only work that adopts zero-centering *per-prompt*. However, their core motivation lies in the identifiability of RMs, which is distant from ours. We build a solid reason for adopting a *batch-wise* sum-to-zero constraint regarding the mitigation of reward model over-optimization. We would happily incorporate the discussion on [2] for our final version of the paper. However, we emphasize that our method remains novel from its theoretical background and empirical support with varying model sizes, types, and datasets.
>
> &nbsp;
>
>
> | Qwen2.5-3B | Gold Reward Mean | Gold Reward Std |
> |:---:|:---:|:--:|
> | RM$\_\text{BT}$     | 0.120  | 0.045  |
> | RM$\_\text{BT-BSR}$ | **0.123**   |  0.045  |
> > **Table A**. Qwen2.5-3B-SFT trained with RLOO with each reward model for 5 epochs, evaluated by the gold reward model (ArmoRM). The reward scores are NOT normalized, unlike Figure 5. Unnormalized score of ArmoRM typically ranges around 0.1-0.2 [3]
>
>
> &nbsp;
>
> **Reference**
>
> [1] Lambert, Nathan, et al. "Rewardbench: Evaluating reward models for language modeling." arXiv preprint arXiv:2403.13787 (2024).
>
> [2] Eisenstein, Jacob, et al. "Helping or Herding? Reward Model Ensembles Mitigate but do not Eliminate Reward Hacking." First Conference on Language Modeling.
>
> [3] Wang, Haoxiang, et al. "Interpretable Preferences via Multi-Objective Reward Modeling and Mixture-of-Experts." Findings of the Association for Computational Linguistics: EMNLP 2024. 2024.

---

### Official Review · Reviewer_rCnJ · 2025-03-15

**Overall Recommendation:** 2

**Summary:**

The paper investigates the cause of reward model over-optimization in RLHF and finds that it stems from the increasing variance of the final-layer outputs (hidden states) in the reward model (RM). The authors propose Batch-wise Sum-to-Zero Regularization (BSR) for RM training, which penalizes the second moment of rewards at the batch level. Experiments show that BSR improves RM stability and enhances LLM fine-tuning via RLOO.

**Claims And Evidence:**

The claims are generally well-supported, but some concerns remain:

* The theoretical discussion of BSR in Section 4.2 is unclear. For example, in lines 265–269, the notation $\prec$ is ambiguous. Does it mean $\leq$? If so, considering cases where $r$ is large, should the inequality direction be reversed?

* A misalignment between the hypothesis in Section 4.1 and the proposed method in Section 4.2 appears to exist. Section 4.1 argues that, considering the final-layer computation of $r(x,y) = W h(x,y)$ in the reward model, reward over-optimization occurs due to the increasing variance of $\| h(x,y) \|$ (or $\| h(x,y_w) - h(x,y_l) \|$) while the size of the projection head $W$ remains approximately constant throughout RM training. However, BSR directly penalizes the reward magnitude, which might unnecessarily reduce $W$, contradicting the initial hypothesis. To validate this hypothesis, an additional experiment regularizing $\| h(x,y) \|$ or $\| h(x,y_w) - h(x,y_l) \|$ should be included. Alternatively, if the focus is on BSR itself, Section 4.1 should analyze $r(x,y)$ directly instead of decomposing it into $W$ and $h(x,y)$.

**Essential References Not Discussed:**

To the best of my knowledge, there are no essential references missing.

**Experimental Designs Or Analyses:**

The experimental design is well-structured, and the dataset construction process is appropriate for evaluating the proposed hypothesis and the effect of BSR.

**Methods And Evaluation Criteria:**

The proposed method and evaluation criteria are appropriate for the problem. The use of RM-Bench and Length-Controlled AlpacaEval 2.0 effectively assesses the impact of BSR on reward model stability and RLHF fine-tuning.

**Other Comments Or Suggestions:**

- To better validate the role of BSR in addressing the hypothesis, it would be helpful to show how Figures 1 and 2 change with and without BSR.
- Line 135: Should $D_{\text{KL}}$ be marginalized over $x$?
- Line 185: Should "softmax value" be "reward value"?

**Other Strengths And Weaknesses:**

One concern is that the performance improvement from the proposed method appears marginal in some evaluations, such as Gold RM evaluation (Figure 5) and RM-Bench (Table 2). The results seem sensitive to hyperparameters and minor experimental settings, raising the possibility that the observed gains could be reversed under slightly different conditions.

**Questions For Authors:**

Table 2 shows a noticeable drop in Easy Acc in RM-Bench when applying BSR. How should this be interpreted? Is there an underlying reason why BSR negatively impacts performance on easier tasks?

**Relation To Broader Scientific Literature:**

This paper is related to the broader literature on LLM alignment, particularly in addressing reward hacking and reward model over-optimization in RLHF.

**Theoretical Claims:**

The paper does not present a formal theoretical analysis, but its mathematical arguments raise concerns about ambiguity. For instance, the claim of "unconstrained hidden state norm dispersion" may be overstated. In ordinary models (excluding tabular models), the sigmoid function term in the gradient of the BT model should reduce the gradient magnitude as the hidden state norm dispersion increases. Moreover, Figure 2 suggests that the hidden state norm dispersion saturates early.

---

> ### Author Rebuttal · Authors · 2025-04-01
>
> We appreciate the reviewer’s detailed comments. Below, we address each concern:
>
>
> 1. **Claims/Evidence #1 – Notations in Section 4.2**:  We regret the confusion from the ambiguous explanations in lines 261–271 and acknowledge that lines 265–269 apply only to a specific case. In the final version, we will include a concise analysis of  $\frac{\partial \mathcal{L}\_\mathrm{BSR}}{\partial h(x,y)}$, showing that it introduces a linear penalty on the growing reward scale, which indirectly limits the expansion of $\|h(x,y)\|$. Thus, $\mathcal{L}\_\mathrm{BSR}$ properly complements the BT loss with regularization.
>
> 2. **Claims/Evidence #2 – Is BSR properly regularizing the hidden state norms?**:  Logit normalization (i.e., $\mathcal{L}\_\text{BT-Norm}$ in Section 4.3) decouples the reward margin from the hidden state norm by dividing the reward by its L2 norm, making the loss scale-invariant [1]. Our analysis in Section 4.1 shows that the reward scale is largely driven by the hidden state norm since $||W_p|| \simeq 1$. However, Figure 4 demonstrates that the strictly normalized loss (RM$_\text{BT-Norm}$) performs substantially worse, suggesting that completely discarding magnitude information removes a valuable discriminative signal for out-of-distribution generalization. In contrast, our BSR method softly regularizes the norm, mitigating over-optimization while preserving useful information. Moreover, Table A shows that even with BSR, the projection head’s norm remains around 1, confirming that the BSR penalty is effectively propagated without fully suppressing the norms.
>
> 3. **Theoretical Claims #1 – Ambiguity in “unconstrained norm dispersion”**: We acknowledge that “unconstrained” may not be ideal, given the sigmoid function’s gradient mechanism. Nonetheless, without a proper regularizer like BSR, norm dispersion can be amplified, especially under certain hyperparameter choices (e.g., learning rate), and we have fully demonstrated this throughout the paper. In the final version, we will revise the term to “excessive norm dispersion” or similar.
>
> 4. **Strengths and Weaknesses #1 – Generalizability of the Method**:  We emphasize that the ultimate evaluation of reward models (RMs) is based on their performance in the RLHF stage. As shown in [2], downstream performance varies significantly with the RM used. Despite marginal improvements in RM benchmarks in some cases, BT-BSR outperforms in the RLHF stage while mitigating verbosity bias [3]—as evidenced by shorter responses and a higher win rate (Table 3). Combined with the improvements seen in RM benchmarks (Figure 4 and Table 2), our regularization method is both generalizable and practically valuable.
>
> 5. **Comments or Suggestions #1 – Further Validation for Figures 1 and 2 with BT-BSR**: Table A confirms that $\|W_p\| \simeq 1$ for BT-BSR checkpoints (as shown in Figure 1). Figure 2 empirically supports that the BT loss triggers norm dispersion, while the comparison between Figures 3(a) and 3(b) confirms that BSR effectively mitigates this phenomenon.
>
> 6. **Comments or Suggestions #2 – Questions on Notations**: Since $\mathbb{D}\_\mathrm{KL}$ is conditioned on the prompt $x$, both the reward and the KL penalty should be under $x \sim \mathcal{D}$. Additionally, lines 184–188 bridge the over-confidence issue in classic multi-class classifiers to RMs with shared projection heads. (For further details, please refer to our comment on “Q2” for reviewer YT72.) We will clarify these notations in the final version.
>
> 7. **Question #1 – Drop in Easy Tasks**: Appendix O of RM-Bench [4] shows that “hard accuracy” has the highest correlation with policy performance ($r=0.45$) while “easy accuracy” is near 0 ($r=0.07$). Although the drop in “easy accuracy” may stem from a compressed representation space due to BSR, it is crucial that “hard accuracy” aligns with actual policy performance after RLHF. Our experiments in Figure 5 and Table 3 confirm that BT-BSR improves outcomes in both controlled and practical settings.
>
> &nbsp;
>
> |   | **$\| W_p \|$** |
> |:--:|:--:|
> | **Qwen2.5 (1.5B)** |  0.9884 |
> |  **Qwen2.5 (3B)**  |  0.9933 |
> | **Llama-3.2 (1B)** |  0.9896 |
> | **Llama-3.2 (3B)** |  0.9905 |
>
> > **Table A**: The norm of the projection head for four different models after reward modeling with BT-BSR.
>
> &nbsp;
>
> **Reference**
>
> [1] Wei, Hongxin, et al. "Mitigating neural network overconfidence with logit normalization." International Conference on Machine Learning. PMLR, 2022.
>
> [2] Meng, Yu, Mengzhou Xia, and Danqi Chen. "Simpo: Simple preference optimization with a reference-free reward." Advances in Neural Information Processing Systems 37 (2024): 124198–124235.
>
> [3] Yann Dubois, et al. “Length-controlled alpacaeval: A simple way to debias automatic evaluators.” First Conference on Language Modeling.
>
> [4] Liu, Yantao, et al. "RM-Bench: Benchmarking Reward Models of Language Models with Subtlety and Style." The Thirteenth International Conference on Learning Representations.

---

### Decision · Program_Chairs · 2025-05-01

**Decision:**

Accept (poster)

**Comment:**

The reviewers generally ranged from weakly in favor of acceptance to outright rejection, which upon reading the paper I struggle to fully agree with, but nor do I wish to ignore them.

The two primary concerns (novelty and significance of performance gains) are certainly reasonable concerns. BSR *looks* like many other methods, at least at a cursory level. Normalization of rewards and advantages is fairly common (at the policy learning stage), and regularizing everything from weights to activations with a squared loss is a reliable technique. However, the author(s) do more than simply throw that reliable technique against a new wall. It is well motivated and those motivations empirically investigated, and alternatives evaluated. In terms of the significance of the improvements on reward modeling and eventual policy learning, again I agree there is reason for concern or at least caution.  However, the empirical work shows a modest but generally consistent benefit.

As such I see these two concerns as less worrisome than perhaps some of the reviewers, and when coupled with a (to my reading) clear overall narrative, organization and experimental design make this a paper with contributions worth sharing.